# OpenReview forum: "Dialogue Action Tokens: Steering Language Models in Goal-Directed Dialogue with a Multi-Turn Planner"
_ICLR.cc/2025/Conference — Submitted to ICLR 2025_

### Official Review · Reviewer_RkAj · 2024-10-22

**Soundness:** 2
**Presentation:** 2
**Contribution:** 2
**Rating:** 5
**Confidence:** 4

**Summary:**

This work proposes a light-weight RL-based technique to improve language model's goal completion ability. The authors propose Dialogue Action Tokens (DAT), which are prefix tokens emitted by a small (trained) planner model to steer a larger LM's response at test time. During inference time, DAT uses the feature vector from the context's last word from the larger frozen LM, feeds them into a small MLP model (the planner) to output L-tokens, and finally prefixes the input texts with these L-tokens and prompt the frozen LLM for generation. During training, only the small planner model is trained using policy gradient. The author then evaluated DAT in a social intelligence benchmark (Sotopia) and a red-teaming benchmark (HarmBench), and showed improved performance compared to alternatives such as no training.

**Strengths:**

- Since DAT only optimizes a small planner model, it is a compute-efficient approach to improve performance of LLMs.
- The authors presented diverse experiments in social intelligence benchmarks (Sotopia) as well as red-teaming benchmarks (HarmBench) to validate the effectiveness of their approach

**Weaknesses:**

1. I believe there is a significant amount of related work being overlooked. This work proposes DAT motivated by the lack of RL techniques in optimizing utterance-level MDP (L31-39, L128-131) as well as planning/optimizing in continuous action space (L47-49), which is not true. Optimizing utterance-level MDP has been extensively explored by work including but not limited to NLPO [1], RvS [2], ILQL [3], and more. There is little to no mention to these methods, and there is no comparisons to direct RL training approaches in the experiments. Additionally, planning/optimizing in continuous vector space has also been explored by many prior work, especially in task oriented dialogues. For example, LAVA [4] and TCUP [5] optimizes the LM in a latent action space by formulating the auto-regressive generation process as variational inference, and uses RL algorithms such as policy gradient for direct optimization. These were also of high relevance but not mentioned or compared against.

2. Some design choices about DAT seems questionable. For example, the authors only uses a very small planner model, and only L=2 prefix tokens is used to steer the LLM output (presumably to prevent language degradation mentioned in the introduction). However, this presents a significant trade-off between performance v.s. robustness, as increasing L may again cause language degradation. Has the authors experimented with L>2?

3. Comparisons made in the experiments in Section 7 Sotopia were not fair. The author compared a DAT-trained model against no-training/directly prompting LLaMA-2, LLaMA-3, or GPT-4. However, DAT should be compared against *many existing RL training methods including [1-5] and even PPO [6]*, since Sotopia tasks can be formulated as an MDP problem.

4. Comparisons made in the experiments in Section 8 Red Teaming were not fair. 1) Baselines include GDG and PAIR, which constructs an adversarial prompt by searching prefixes/prompts *at test time*, whereas DAT additionally *trained the planner against the defender* before testing. Similar to section 7, there should be comparisons against direct RL methods. 2) DAT performance became *significantly worse against a much weaker defender (LLaMA-2-7b-chat)*, whereas other baselines show improved performance. Is this an indication of reward over-optimization of DAT-training?

---

References:

[1] Ramamurthy, Rajkumar et al. “Is Reinforcement Learning (Not) for Natural Language Processing?: Benchmarks, Baselines, and Building Blocks for Natural Language Policy Optimization.” ArXiv abs/2210.01241 (2022): n. pag.

[2] Emmons, Scott et al. “RvS: What is Essential for Offline RL via Supervised Learning?” ArXiv abs/2112.10751 (2021): n. pag.

[3] Snell, Charles Burton et al. “Offline RL for Natural Language Generation with Implicit Language Q Learning.” ArXiv abs/2206.11871 (2022): n. pag.

[4] Lubis, Nurul et al. “LAVA: Latent Action Spaces via Variational Auto-encoding for Dialogue Policy Optimization.” ArXiv abs/2011.09378 (2020): n. pag.

[5] Vlastelica, Marin et al. “Taming Continuous Posteriors for Latent Variational Dialogue Policies.” AAAI Conference on Artificial Intelligence (2022).

[6] Schulman, John, et al. "Proximal policy optimization algorithms." arXiv preprint arXiv:1707.06347 (2017).

**Questions:**

In general, I believe this paper is poorly structured. For example,

- Many empirical details such as "Notes on architecture" (L168-173); "Remarks" (L235-237) should not be mentioned in the method section but rather in experimental setups.
- Experiments section should be sectioned into separate sections that introduces the benchmark, baselines, specific implementation details, results, and ablations/analysis. Currently, there are only two subsections in Section 6 and Section 7 that discusses all the relevant information mentioned above.

---

> ### Author Response · Authors · 2024-11-25
>
> We thank reviewer for their evaluation of our work. Please see our reply below.
>
> **Lack of novelty**
>
> We will revise texts around L31-39, L128-131, and L47-49 to be more precise. And thanks for bringing [1-5] to our attention, we will discuss them in detail in related works. However, by reading these papers in detail, we believe that they are operating in a token-level rather than utterance-level MDP, leading to different technical difficulties and experiment set-up.
>
> **Number of control tokens**
>
> We experimented with different numbers of control tokens in Appendix C (Figure 5).
>
> **Lack of comparisons on Sotopia experiment**
>
> We looked into the suggested baselines but were not sure how they could be applied to our experiment setting.
>
> For [1,3,4,5], all of the proposed techniques operate on token-level MDP for one-pass generation; while DAT is proposed to handle utterance-level planning in interactive dialogue setting.
>
> For [2,6], they are pure RL paper, without mentioning language-related tasks.
>
> We beat the strongest model (GPT-4) from the original Sotopia paper. Given the obvious difference between GPT-4 and Llama2 in parameter number and general capabilities, we believe this is an interesting result for others in the field to know about.
>
> **Unfair comparisons in red-teaming experiments**
>
> We appreciate the reviewer's attention to the fairness of experiment. However, we maintain that ours is fair after double-checking.
>
> Let’s recap the experiment process of GCG-multi and PAIR. To start, there is only one set of prompts (jailbreaking requests) without train/val/test split. GCG-multi trained one single prefix to jailbreak as many of them as possible by eliciting “Sure, here is…” PAIR manually engineered a prompting scheme *with* access to this set. In light of these, we still think DAT is benchmarked fairly.
>
> Both GCG(-multi) and PAIR are model-dependent, meaning they will do separate training for different defender models. However, The DAT model is only trained once against LLaMA3-8B, and tested out-of-the-box against LLaMA2-7B. In this sense, the experiment set-up was in this sense in favor of GCG and PAIR .

---

> ### Comment · Reviewer_RkAj · 2024-11-26
>
> Thank you for your replies. In general, I still believe that many prior work in RL is applicable to the formulation described in this paper, yet they are omitted and not compared against in this work. I detail these reasoning below.
>
> However, I am happy to change my stance if the authors can provide reasons why these prior work mentioned are not applicable in this work, and I am willing to adjust my scores accordingly if it is the case.
>
> ---
>
> > I believe there is a significant amount of related work being overlooked. This work proposes DAT motivated by the lack of RL techniques in optimizing utterance-level MDP (L31-39, L128-131) as well as planning/optimizing in continuous action space (L47-49), which is not true...
>
> Sorry for the confusion, and let me add more context to my previous statement. Works such as NLPO [1] introduces a modified PPO objective that are friendly to perform RL with language models. I believe this already in collision with your statement on L31-33: "This work aims to address this gap by proposing a generic reinforcement learning (RL) technique tailored for LM agents," as many of these work already exist [1-5] and there is no comparison to them.
>
> This work considers "each utterance as a deliberate action" and applies "RL to improve LM agents in goal-directed dialogues" (L40-41). I believe this case is already taken into account by work such as NLPO [1]. Although policy gradient methods like NLPO provides gradient update to *every token* (i.e., token level), in practice the *only reward that is given is on a sequence level*, (i.e., utterance level, see the last three lines on page 3 of [1]). Implementation-wise, this notion of optimizing for an utterance level objective can be seen in Algorithm 1 which 1) which collects response *trajectories*, annotate then with utterance-level rewards (see their Section 3.2), and performs gradient updates to steer the entire policy towards that reward.
>
> In general, I believe token-level RL algorithms are *extension* of sequence level RL to take advantage of token-level rewards, which may be available in some domains. In the context of this work, I believe prior work [1-5] can be easily "converted back" to sequence/utterance level by, for example, specifying zero for all token reward and only assign a terminal reward for the entire utterance (which is common in task-oriented dialogues).
>
> Perhaps I misunderstood your setup, but could you elaborate why there would be "different technical difficulties" in directly applying these techniques in your work?
>
> > Comparisons made in the experiments in Section 8 Red Teaming were not fair. 1) Baselines include GDG and PAIR, which constructs an adversarial prompt by searching prefixes/prompts at test time, whereas DAT additionally trained the planner against the defender before testing.
>
> I believe there is a difference in the notion of "training" we mentioned. In my original question, I refer to training as essentially changing model weights via backpropagation (e.g., either after SFT or RL). In this work, this form of training is done as mentioned in Section 5.1-5.2. In GCG, there is no such gradient updates to any LM but a *search*. Since there is no gradient updates at all, this making GCG lightweight and memory efficient (i.e., compared to model training).
>
> "However, The DAT model is only trained once against LLaMA3-8B, and tested out-of-the-box against LLaMA2-7B". I agree that this is less in favor compared to algorithms such as GCG which (I assume) can access LLaMA2-7B as well. **Could you provide the training time and memory required required for DAT and similarly the time/memory needed for GCG (and other baselines) to optimize a set of prompt?** This should perhaps better clarify whether or not one method uses more resources than others.

---

> > ### Author Response · Authors · 2024-11-26
> >
> > **Re. Sotopia experiment**
> >
> > Thanks for clarifying your review. We acknowledge that our description of the existing literature is inaccurate (L31-39, L128-131, and L47-49) and we will change them in an updated version. Apologize again and thank you for contributing to this work by pointing out.
> >
> > On required comparison with [1-5]:
> >
> > Token-level vs utterance-level is the difference in techniques. However, a deeper difference is that, compared to the one-pass setting used in NLPO (single-player game), our work operates in multi-round conversation (two-player game). From both a game theory or reinforcement learning perspective, the gap between the two has never been small. Game theory concepts like optimization frameworks and simple MDPs break down, requiring Nash equilibria or minimax solutions; RL algorithms like Q-learning, DQN, and basic policy gradients fail due to non-stationarity and adversarial dynamics, necessitating MARL, self-play, or equilibrium-based strategies.
> >
> > Due to these added complexities in multi-round dialogue planning, we gave up token-level planning and focus on utterance-level planning. This poses significantly less computation burden on us. Take NLPO as an example, it requires finetuning the language model, while DAT only needs to train an MLP model with offline data.
> >
> > Our current understanding is that DAT is not competing against token-level RL techniques, but rather works in an orthogonal research field. In a sense, DAT enjoys the lightweight steering by counting on the token-level policy learned within the model weights. Please let us know if this makes sense to you!
> >
> > As a side note, we tested two existing methods suggested by reviewer diQR—PPDPP and GPT-Zero, neither of which requires training access to the LM. We hope this better gourds our work in the existing literature.
> >
> > **Re. Red-teaming experiment**
> >
> > We just tested again—GCG-multi took about 4 hours to converge on HarmBench. GCG requires gradients to flow through the frozen LM to its added parameters/prefix. In fact, it’s called Greedy Coordinate _Gradient_. For the sake of this, GCG poses a higher requirement on max GPU memory size than DAT.
> >
> > The profile of DAT’s computing requirement is different, thus hard to compare directly. Specifically, we need to generate a buffer of dialogues, which calls for a lot of inference computation. The buffer size we used in our experiment is 120,000, but it’s easily parallelizable on clusters. The real RL training of DAT is very fast and cheap in comparison, taking less than 4G GPU memory and can be done within 10 minutes. We want to reiterate that the whole LM is frozen during DAT training; only the planner (an MLP) is trained.

---

> > > ### Comment · Reviewer_RkAj · 2024-11-27
> > >
> > > > I believe there is a significant amount of related work being overlooked. This work proposes DAT motivated by the lack of RL techniques in optimizing utterance-level MDP (L31-39, L128-131) as well as planning/optimizing in continuous action space (L47-49), which is not true...
> > >
> > > I believe you are introducing more variables into our discussion that are independent of the technique of token-level reward/RL v.s. utterance level reward/RL. I still believe prior work such as NLPO [1-5] does not interfere with the setting you explained. I detail these reasons below. Please add clarifications/additional responses in case I missed something.
> > >
> > >
> > > - You mentioned "single-player game" v.s. "our work operates in multi-round conversation (two-player game)". I agree that optimizing towards moving targets (e.g., both party in a two-player game gets parameter update), existing methods such as NLPO [1] may not converge as you mentioned. **However, I believe in this work the second player is always frozen** (e.g., "apply DAT to steer one of the two LM agents so that its social capability gets improved" in Section 6, and "we apply DAT to the attacker LM" in Section 7), so it is essentially a still a single player game. In this setting, I believe all these prior RL work I mentioned applies, and there is no "non-stationarity and adversarial dynamics".
> > >
> > > - "Take NLPO as an example, it requires finetuning the language model, while DAT only needs to train an MLP model with offline data." I believe "requires finetuning the LM" and "train an MLP" is only a very minor difference. Additionally, in this work the goal of DAT was to train a planner model to optimize reward provided by a judge model (L214-215). As mentioned before, since learning is revolved around training a planner model to optimize to maximize this reward, I do not see why work such as NLPO [1] is not applicable. Why can you not train "an MLP model" using NLPO under the same judge's reward? Furthermore, if goal is to improve model A's performance (e.g., attack success) when interacting with another frozen model B, then methods such as NLPO [1] can be directly applied to optimize model A, and would constitute a very intuitive baseline given the experiments in this paper.
> > >
> > >
> > > > Comparisons made in the experiments in Section 8 Red Teaming were not fair. 1) Baselines include GDG and PAIR, which constructs an adversarial prompt by searching prefixes/prompts at test time, whereas DAT additionally trained the planner against the defender before testing.
> > >
> > > "GCG requires gradients to flow through the frozen LM to its added parameters/prefix. In fact, it’s called Greedy Coordinate Gradient." Yes, but just to re-iterate, my original statement is:
> > >
> > > ```
> > > I refer to training as essentially changing model weights via backpropagation (e.g., either after SFT or RL). In this work, this form of training is done as mentioned in Section 5.1-5.2. In GCG, there is no such gradient updates to any LM but a search... Since there is no gradient updates at all, this making GCG lightweight and memory efficient (i.e., compared to model training).
> > > ```
> > >
> > > I agree that gradient is used as a metric to search for the next best candidate tokens in GCG, but there is no *changing weights* in the LM via backpropagation. I thus believe my point was still valid, unless in your implementation of GCG you are updating the model weights?
> > >
> > > However, I appreciate the provided statistics on GPU compute and time. I believe "GCG poses a higher requirement on max GPU memory size than DAT" is because DAT trains a smaller, specialized planner, yet GCG searches using the larger model (despite no gradient updates). I believe this resolves my concern about resource usages, and I have increased my soundness score to 2. I believe the primary concern was still the omission of existing RL techniques as mentioned in my previous response. I will holistically change my scores if that can be addressed.

---

> > > > ### Author Response · Authors · 2024-11-27
> > > >
> > > > I see your point! By taking the added MLP as a new way of finetuning, it's indeed easy to carry out experiment for NLPO and PPO. I tested them out, and the new Table1 will look like below. We wish this makes our work better grounded in the literature. And we will surely revise the texts you pointed out L31-39, L128-131, and L47-49.
> > > >
> > > > |                                | Llama-2-7B-chat       | Llama-3-8B-instruct       |
> > > > |--------------------------------|-----------------------|---------------------------|
> > > > | Llama-2-7B-chat               | $3.24 \pm 0.14$       | $3.25 \pm 0.14$           |
> > > > | Llama-3-8B-instruct           | $3.25 \pm 0.14$       | $3.39 \pm 0.11$           |
> > > > | GPT-4                         | $3.53 \pm 0.14$       | $3.65 \pm 0.12$           |
> > > > | Llama-2-7B-chat w/ PPDPP [Deng et al., 2023] | $3.50 \pm 0.13$       | $3.61 \pm 0.15$           |
> > > > | Llama-2-7B-chat w/ GPT-Zero [Yu et al., 2023] | $3.47 \pm 0.19$       | $3.50 \pm 0.21$           |
> > > > | Llama-2-7B-chat w/ NLPO [Ramamurthy et al.] | $3.55 \pm 0.14$       | $3.67 \pm 0.15$           |
> > > > | Llama-2-7B-chat w/ PPO [Schulman et al.]    | $3.43 \pm 0.15$       | $3.35 \pm 0.16$           |
> > > > | Llama-2-7B-chat w/ self-clone | $3.24 \pm 0.15$       | $3.29 \pm 0.14$           |
> > > > | Llama-2-7B-chat w/ DAT   | **$3.59 \pm 0.13$**   | **$3.73 \pm 0.14$**       |
> > > >
> > > > For your question on red-teaming experiment fairness, it eventually hinges on what you think is **training**. In my opinion, if GCG uses gradient to train some randomly initialized parameters (prefix embeddings) to optimize ASR, it's a fair comparison to us using gradient to train an MLP for the same goal.

---

> ### Comment · Reviewer_RkAj · 2024-11-27
>
> I appreciate the additional results on baselines such as NLPO [1]. I just noticed that this issue of missing many related/prior work in this paper is also mentioned by Reviewer diQR. I raised my score to 5 to reflect the authors effort in conducting these additional experiments, but I am hesitant to raise it further because this issue of missing many related work would also entail substantial re-writing/re-position of the paper. It is unclear to me if this would be appropriate in so far as this conference submission is concerned. I leave this judgment to meta-reviewer/AC of this conference.

---

### Official Review · Reviewer_sFxb · 2024-10-23

**Soundness:** 3
**Presentation:** 3
**Contribution:** 4
**Rating:** 8
**Confidence:** 3

**Summary:**

This paper presents a novel approach, Dialogue Action Tokens (DAT), to improve the long-term performance of Large Language Models (LLMs) in goal-oriented dialogues. The proposed method introduces a planner module and an up-mapping matrix to backbone LLMs, enabling explicit prediction of dialogue actions at each turn. These predicted action embeddings are prepended to the dialogue history when generating responses. Both modules can be trained in continuous vector space using Reinforcement Learning (RL) with a dedicated reward function. The training process consists of two steps: (1) cloning the original backbone LLM's behavior, which serves as pretraining for the two modules; and (2) fine-tuning the planner module with the reward function while freezing the up-mapping matrix and backbone LLM. The method is evaluated on two tasks, Social Capability and Red Teaming, demonstrating significant improvements over strong baselines.

**Strengths:**

1. The idea of explicitly prepending dialogue action tokens to the dialogue history is novel and interesting, offering a valuable contribution to future research in the community.
2. Instead of relying on prompt engineering, the proposed method introduces lightweight, trainable parameters to LLMs, which can be trained through RL. This approach makes the training process more mathematically sound and effective.
3. The proposed method achieves impressive gains on both tasks compared to state-of-the-art results, particularly on the Red Teaming task. The results are convincing, supported by extensive experiments and ablation studies.
4. The method has good theoretical generalizability, potentially applicable not only to goal-oriented dialogue but also to non-goal-oriented scenarios if the underlying purpose can be defined. Additionally, the technique is not limited to specific backbone models or RL algorithms.

**Weaknesses:**

1. A notable limitation is that the proposed method requires a reward function for RL, which can be hard or costly to acquire as mentioned in the paper. This may restrict the use of DAT in certain scenarios where reward signals are difficult to obtain.
2. While the evaluation results are promising, they rely solely on automatic methods. Incorporating human evaluation would provide additional validation and make the results even more convincing.

**Questions:**

1. Are predicted action tokens at each turn also added to the dialogue history?
2. In Table 3,  why does Llama-2-7B have better performance than Llama-3-8B agaist two of single-turn attackers?

---

> ### Author Response · Authors · 2024-11-25
>
> We thank reviewer for their evaluation of our work. Please see our reply below.
>
> **Usage of powerful reward models**
>
> We agree that this is a big assumption of our work. We see two interesting directions here: (1) building cheap replacement for API calls, which will accelerate as well as reduce the cost of DAT training (2) building better reward models, which has been explored under the agenda of human preference elicitation.
>
> We will further highlight this limitation in the write-up.
>
> **Validity of reward function**
>
> Our work builds on the assumption that the LLM judges provided by Sotopia and HarmBench are correlated with humans as a cheap proxy. Indeed, both papers have investigated this issue extensively with positive answers (section 5.2 in [sotopia](https://arxiv.org/pdf/2310.11667), and table 3 in [harmbench](https://arxiv.org/pdf/2402.04249)). We will add reference to them in prose.
>
> By reading dialogue under and before DAT optimization (e.g., those in the appendix), I could qualitatively observe that the reward function is order-consistent with how I would personally rank the dialogues. However, we acknowledge that a systematic human study could confirm the validity of the LLM judges.
>
> **Questions**
>
> + No, the action tokens are not added into dialogue history.
> + For defenders, the ASR is the lower the better. Table 3 says LLaMA2-7B is worse at defending against GCG than LLaMA3-8B; but better against PAIR and DAT attack. Note that the DAT attack is optimized directly against LLaMA3-8B so the difference is expected.

---

> > ### Comment · Reviewer_sFxb · 2024-11-26
> >
> > Thanks for the further clarification. After reading the discussion, I will remain my rating unchanged.

---

### Official Review · Reviewer_LBPv · 2024-11-05

**Soundness:** 3
**Presentation:** 3
**Contribution:** 3
**Rating:** 8
**Confidence:** 4

**Summary:**

Whilst large language models have illustrated impressive performance in many tasks, it has also become clear that they often struggle to plan sufficiently when faced with longer term or challenging goals. This work proposes the use of simplified dialogue action tokens used to steer the LLM in the right direction. They use a separate, small, planning module which assist the LLM in planning for longer term goals. Through RL, this planner learns to successfully steer the LLM through a complex goal, such as those present in the Sotopia scenarios.

**Strengths:**

- This work presents a relevant method for improving the long term planning of large language models using dialogue action tokens.
- The method is well tested in the two scenarios, Sotopia and the Red Teaming Scenarios.
- A small, and hence "cheap", planning module to insert tokens for planning improves planning performance significantly. Hence, providing a computationally efficient method to improve the performance of LLMs in long term tasks without expensive LLM fine-tuning.

**Weaknesses:**

Whilst the small planning module does successfully steer the LLM, it does not improve the actual general planning ability of the LLM. Further, through the bottleneck of the DAT tokens, information can be lost which could have been beneficial for generating better responses. This leaves whether the LLM could learn to perform this planning unanswered.

**Questions:**

Have you tested whether RL to generate such planning tokens directly by the LLM can be done? Possibly using a penalty to avoid diverging from natural language.

---

> ### Author Response · Authors · 2024-11-25
>
> We thank reviewer for their evaluation of our work. Please see our reply below.
>
> **Improve LLM planning without external modules**
>
> Appreciate this thought-provoking question. I haven’t tested whether it’s possible to distill the planner into the base language model. It’s a promising future direction. A neural network smaller than common LM can reach grand-master level planning in chess ([paper](https://arxiv.org/abs/2402.04494)), so we shouldn’t rule out the possibility of an LLM planning well internally. DAT potentially offers a scaffold to advance the internal planning capability of LLMs.

---

### Official Review · Reviewer_jzcM · 2024-11-05

**Soundness:** 3
**Presentation:** 3
**Contribution:** 3
**Rating:** 6
**Confidence:** 3

**Summary:**

The paper looks at training an LLM to perform the role of a conversational agent which has certain goals, e.g. steering a user towards a certain response. The paper approaches this by training a small module which produces essentialy control tokens (2 in the paper) that are the same dimensions as the LLM token embeddings. The LLM is the conditioned on these extra tokens when producing a response. The paper gives a good description of RL and past work, and motivates taking this approach primarily as it doesn't change the LLM, so by design doesn't degrade its core language abilities.
This has been done before under terms like prompt learning, with differences mainly in how the "control tokens" are obtained.

**Strengths:**

Clear motivation, tested on open data and compared against prior methods. Results indicate that the proposed method is working, although there are a few asterisks on the evaluation

**Weaknesses:**

* The evaluation is based on the same signal being optimised by the reported model. This is a bias towards the proposed approach being better than the others. Some what tricky to avoid, but human evaluation would be one (admittedly expensive) approach to more robust evaluation.
* The method relies on a very powerful evaluator. This is ok, but important to note (as the paper does). However there is no reporting here of how accurate even the prompted GPT was on this task. I presume such was measured as part of prompt engineering your way to a good LLM judge? It would be helpful to know how accurate the evaluator was, both for interpretation of the actual reported results, but also for how much signal (versus noise) was being input to the RL training.
* are any of those results statistically significant? Looking at table 1 alone, I'm not sure they would be, as most seem to be overlapping when considering the 68% intervals

**Questions:**

* Interested in the authors speculation on why the certain words were obtained as interpretations of the policy model (lines 433-434)?
* Also, the policy only outputs 2 vectors, are all those words tokenised into 2 tokens?
* Should the partner model be a different LLM? Presumably it is significantly easier to learn how to bias "your twin" so to speak, compared to another model? Maybe that's a speculative question worthy of future analysis. Interested in your thoughts.

---

> ### Author Response · Authors · 2024-11-25
>
> We thank reviewer for their evaluation of our work. Please see our reply below.
>
> **Potentially overfitting reward function**
>
> Indeed, compared to other red-teaming methods that optimize against their specific heuristics, DAT runs the risk of hacking for the test evaluator. The point here is that such an evaluator is shown to be robust (see next question) therefore optimizing a given evaluator signal is challenging/meaningful.
>
> **Validity of reward function**
>
> Our work builds on the assumption that the LLM judges provided by Sotopia and HarmBench are correlated with humans as a cheap proxy. In fact, both papers have investigated this issue extensively with positive answers (section 5.2 in [sotopia](https://arxiv.org/pdf/2310.11667), and table 3 in [harmbench](https://arxiv.org/pdf/2402.04249)). We will add reference to them in prose.
>
> By reading dialogue under and before DAT optimization (e.g., those in the appendix), I could qualitatively observe that the reward function is order-consistent with how I would personally rank the dialogues. However, the noise is definitely there, both troubling RL training and obfuscating evaluation to a certain degree. We will mention these in the write-up.
>
> **Statistical significance**
>
> In Table 1, DAT gives 2.5-sigma and 3.4-sigma improvement to LLaMA2-7B, against LLaMA2-7B and LLaMA3-8B respectively. These translate to a p-value of 1.24% and 0.034% respectively, which are lower than the commonly accepted 5%.
>
> **Speculations on decoded control vectors**
>
> Here is my try. In the given examples of nation names and girls’ names, the tokens will shift the model into a friendly and strategic way of talking. Name is especially interesting as it fits the format of the playwright: “name: utterance.”
>
> The predicted control vectors will bypass any decoding/encoding layer, it’s directly fed into the first layer of Transformer, acting just like normal word embeddings.
>
> **Partner parameterized by a different model**
>
> For social capability, ideally the partner will be real humans; for red-teaming, the defender often takes the form of a close-source model. In both cases, the partner is very different from our in-house open-source model. We believe this future line of investigation is important.

---

> > ### Comment · Reviewer_jzcM · 2024-11-27
> > **Acknowledging feedback**
> >
> > Thanks for responding to the review. Whilst it's helpful to cite those works which evaluated/calibrated the reward function, I do still feel that the comparison against other methods is biased given the proposed method only is optimising this function.

---

### Official Review · Reviewer_diQR · 2024-11-06

**Soundness:** 2
**Presentation:** 2
**Contribution:** 2
**Rating:** 5
**Confidence:** 4

**Summary:**

The authors propose an approach called Dialogue Action Tokens which serves as a continuous vector representing actions that an LLM should take throughout the course of a conversation. The proposed approach is evaluated on SOTOPIA, as well as a novel task called ``multi-turn red-teaming'' designed by the authors.

**Strengths:**

Overall, the writing is clear and the problem formulation is sound -- it is reasonable to consider as an MDP as described in Section 3.

The general idea of adapting LLMs to downstream conversational scenarios is also important, and the generalizability of a continuous action vector is also an important concept.

The proposed multi-turn red-teaming task is also interesting.

**Weaknesses:**

My primary concerns with this work are that the baselines are not properly set, and the overall novelty of the proposed work is limited. The work makes a strong claim that LLMs can only be steered for downstream applications via prompt engineering, and that the proposed DAT method addresses the gap by virtue of being an RL-based approach which plans actions across long horizons. However, the work does not engage with the existing literature on multi-turn conversations. In particular, the main experimental results in Table 1/3 do not include comparisons to other dialogue action planning baselines (e.g., [1, 2, 3]). It is also not clear how the proposed work is fundamentally different than existing work looking at continuous representations of dialogue acts (e.g. [4]).

[1] Plug-and-Play Policy Planner for Large Language Model Powered Dialogue Agents, ICLR 2024

[2] Prompt-Based Monte-Carlo Tree Search for Goal-Oriented Dialogue Policy Planning, EMNLP 2023

[3] Planning Like Human: A Dual-process Framework for Dialogue Planning, ACL 2024

[4] DiactTOD: Learning Generalizable Latent Dialogue Acts for Controllable Task-Oriented Dialogue Systems, SIGDIAL 2023

**Questions:**

Why is the reasoning behind proposing red-teaming as a multi-turn dialogue task?

Given that prompt engineering is an effective way to adapt an LLM for downstream conversational tasks (L31), have you compared DAT to any prompting approaches for goal-oriented dialogue? Can DAT be combined with any pre-existing prompting approaches?

This isn't a question, but - I think it would be helpful to examine the literature on mixed-initiative conversational agents, if the authors intend to frame the overall work as a contribution to work on dialogue.

---

> ### Author Response · Authors · 2024-11-25
>
> We thank reviewer for their evaluation of our work. Please see our reply below.
>
> **Lack of novelty**
>
> We thank the reviewer for bringing [1,2,3] to our attention, we acknowledge that the formulation of dialogue as utterance-level MDP and predicting “action” with external RL-trained modules has been explored. Therefore, we will discuss [1,2,3] in related works. However, we want to point out that our predicted “action” is a continuous vector while, in [1,2,3], the action is predicted from a fixed set of template actions. We believe this discrete design obstructs a more flexible/powerful RL-based control. For additional experiment, see next section of this reply.
>
> This leads to [4], which indeed used continuous vectors for control but is still considerably different from DAT. In [4], the “action” is BERT embedding of a slightly more flexible set of texts than in [1,2,3], but still is constrained by manual design of the viable strings. In DAT, there doesn’t exist such mapping from continuous “action” to natural languages, which could unleash the power of RL better in our opinion. Last but not least, [4] is trained with a kind of imitation learning rather than RL.
>
> **Additional experiment**
>
> We tried to implement the mentioned baselines, here are our encountered difficulties and progress so far. Updates will be posted here and/or reflected in the revised manuscript.
>
> For PPDPP [1], Deng et al. first defined a set of “actions” suited for the task of bargaining, then collected a dataset of bargaining dialogue–action pairs for imitation learning. We tried to force fine tuning on this dataset, run RL training and test on Sotopia, but the experiment's results are not yet statistically significantly different from the LLaMA2-7B-chat baseline (3.24 ± 0.14) shown in our Table 1. We think the problems are two-fold. First, the action defined by PPDPP is not best suited for Sotopia which covers more diverse scenarios than bargaining. Second, we lack such an imitation learning dataset to bootstrap RL. We are still working on this by defining a larger set of actions and directly starting RL training.
>
> For GDP-Zero [2], Yu et al. used ChatGPT for all their experiments. We plugged LLaMA2-7B-chat and Sotopia into the codebase. However, our local computing resources are dwarfed by the extensive computing requirement put by the MCTS search. Experiments are running but are rather slow. We will update results in the manuscript after finishing them, but it’s unlikely that we can get it before the rebuttal deadline.
>
> DPDP [3] is contingent on the reproduction of [1,2], we will get onto it afterwards.
>
> **Motivation of red-teaming as multi-turn dialogue**
>
> Since the real-world usage of close-coursed LLMs allows multi-round dialogue, we believe this is a realistic angle of attacking and safety research. For the purpose of this paper, red-teaming also enjoys the advantage of being easy to verify (compared to Sootpia).
>
> If you are interested, recently we have seen several newer attacks that exploit multi-round dialogue for stronger attacks, see [this](https://blog.haizelabs.com/posts/cascade/).
>
> **Combining DAT with prompt engineering**
>
> In Sotopia experiments, we followed the benchmark set-up which already included hand-engineered system prompts; in the red-teaming experiment, we also crafted some system prompts by ourselves (end of page 15). In this sense, we’ve already combined prompt engineering with DAT as the reviewer suggests.
>
> **Expand onto mix-initiative agents**
>
> We will look into this!

---

> > ### Comment · Reviewer_diQR · 2024-11-25
> >
> > Thank you for your response to my review. Given the claims of the paper, I believe it is crucial to demonstrate improved performance over discrete action planning baselines like [1,2,3] and continuous vector-based approaches like [4].
> >
> > > *"Last but not least, [4] is trained with a kind of imitation learning rather than RL."*
> >
> > I believe that demonstrating improvements over a baseline such as [4] would strengthen your claim on significance.
> >
> > >*" First, the action defined by PPDPP is not best suited for Sotopia which covers more diverse scenarios than bargaining. Second, we lack such an imitation learning dataset to bootstrap RL. We are still working on this by defining a larger set of actions and directly starting RL training."*
> >
> > After checking [1] again I would like to clarify that PPDPP has also been evaluated on diverse tasks such as Tutoring Dialogues as well as Emotional Support, rather than only bargaining dialogues. In terms of defining actions for SOTOPIA, it may be helpful to consider the discussion in Section 2.2 in their paper as a starting point.
> >
> > >*"motivation of red-teaming"*
> >
> > Thanks for the helpful link and discussion, I will read further on this!
> >
> > ---
> >
> > At this time, I feel that the paper is incomplete given that the work has not been positioned in the context of existing literature on action planning, and there has been a limited selection of baselines, which as a result impacts in the novelty and significance of this work. The paper has also not currently been updated to account for the discussion in my review or those of the other reviewers. However, I am happy to discuss further should the Authors make further progress/expand upon the points mentioned in their rebuttal.

---

> > > ### Author Response · Authors · 2024-11-26
> > >
> > > We thank you for your suggestions on experiment set-up. We’ve gotten results for [1,2], which will be added to a revised version of the manuscript.
> > >
> > > |                                | Llama-2-7B-chat       | Llama-3-8B-instruct       |
> > > |--------------------------------|-----------------------|---------------------------|
> > > | Llama-2-7B-chat               | $3.24 \pm 0.14$       | $3.25 \pm 0.14$           |
> > > | Llama-3-8B-instruct           | $3.25 \pm 0.14$       | $3.39 \pm 0.11$           |
> > > | GPT-4                         | $3.53 \pm 0.14$       | $3.65 \pm 0.12$           |
> > > | Llama-2-7B-chat w/ PPDPP [Deng et al., 2023] | $3.50 \pm 0.13$       | $3.61 \pm 0.15$           |
> > > | Llama-2-7B-chat w/ GPT-Zero [Yu et al., 2023] | $3.47 \pm 0.19$       | $3.50 \pm 0.21$           |
> > > | Llama-2-7B-chat w/ self-clone | $3.24 \pm 0.15$       | $3.29 \pm 0.14$           |
> > > | Llama-2-7B-chat w/ DAT   | **$3.59 \pm 0.13$**   | **$3.73 \pm 0.14$**       |
> > >
> > >
> > > We are happy to cite and compare to [4]. However, we can’t find any open-source code for reproducing. Could you please let us know how to find it? Otherwise, we find it quite non-trivial to faithfully reproduce DiactTOD on our environments.
> > >
> > > Oh and we also want to mention that, besides existing literature like [1,2], we also beat the strongest model (GPT-4) from the original Sotopia paper. Given the obvious difference between GPT-4 and Llama2 in parameter number and general capabilities, we believe this is an interesting result for others in the field to know about. What do you think?

---

> > > > ### Author Response · Authors · 2024-11-28
> > > >
> > > > Hello!
> > > >
> > > > Please let us know what you are thinking right now. Thanks again for your valuable feedbacks on our manuscript.

---

> > > > > ### Comment · Reviewer_diQR · 2024-12-02
> > > > >
> > > > > Thanks for your detailed response. I appreciate the effort you have put towards running baselines such as PPDPP and GDP-Zero, and I believe they are helpful in strengthening your paper. Please describe your implementation of these baselines (e.g., chosen action spaces) in the revised version of the paper. I have increased my overall score to a 5.
> > > > >
> > > > > > Given the obvious difference between GPT-4 and Llama2 in parameter number and general capabilities, we believe this is an interesting result for others in the field to know about.
> > > > >
> > > > > Indeed this is a nice finding -- this is consistent with other works which find that smaller LLMs are able to outperform larger ones with more specialized tuning.
> > > > >
> > > > > > We are happy to cite and compare to [4]. However, we can’t find any open-source code for reproducing. Could you please let us know how to find it? Otherwise, we find it quite non-trivial to faithfully reproduce DiactTOD on our environments.
> > > > >
> > > > > It is an important baseline as they directly propose a similar idea and ideally these experiments should be investigated prior to publication. I am not sure if the implementation has been released, but you could try asking the authors if they have the code for it, or otherwise try to reproduce it directly after the discussion period?
> > > > >
> > > > > ---
> > > > >
> > > > > Beyond these points, as mentioned by Reviewer LBPv I think it would be important to further investigate how DAT actually influences planning. As per my initial review, I also believe it would be helpful to better situate this work in relation to existing approaches towards dialogue action planning.

---

### Meta-Review · Area_Chair_ARw1 · 2024-12-17

**Metareview:**

The paper introduces Dialogue Action Tokens (DAT), a lightweight RL-based method designed to improve the goal-directed capabilities of LLMs in dialogue tasks. By using a small planner module to generate action tokens, which are prepended to the dialogue input, the approach steers the LLM without altering its core parameters, preserving its foundational abilities. Evaluated on Sotopia and a novel multi-turn red-teaming task, DAT demonstrates enhanced performance over strong baselines, particularly in long-term planning and complex goal-oriented scenarios.

Strengths:

1. The approach is compute-efficient, as it optimizes only a small planner model to improve LLM performance.
2. The multi-turn red-teaming task is interesting and effectively demonstrates the method's applicability.
3. The experiments are diverse, spanning both social intelligence benchmarks (Sotopia) and red-teaming benchmarks (HarmBench).
4. The method employs trainable parameters prepended as tokens, providing a practical alternative to prompt engineering and expensive fine-tuning, though two reviewers have concerns about the novelty and impact of this work.

Weaknesses:

1. Novelty: The submission lacks sufficient comparison with existing literature on multi-turn conversations and related long-horizon planning methods, raising concerns about the originality of the proposed approach. Furthermore, the authors do not clarify how their method fundamentally differs from existing work on continuous representations of dialogue acts. The submission also overlooks a substantial body of prior research on optimizing utterance-level MDPs and planning in continuous vector spaces, which undermines the claims of novelty. Specifically, RL-based methods such as NLPO, RvS, and ILQL have already addressed utterance-level MDP optimization, while approaches like LAVA and TCUP have explored continuous action spaces in task-oriented dialogues.

2. Experiments: The experiments do not include comparisons to relevant dialogue action planning baselines, which weakens the evaluation of the proposed method. Additionally, the comparisons in Section 7 are unfair, as the proposed method is evaluated against prompting-based baselines rather than RL-based methods, which are more appropriate for tasks formulated as MDP problems.

3. Evaluation: The reliance on automatic evaluation introduces potential biases, especially since the evaluation signal directly aligns with the model's optimization objective. Human evaluations, while costly, would add robustness to the reported results and help address concerns about evaluator accuracy and statistical significance.

4. Design choices: The paper lacks sufficient justification for certain design decisions, such as the use of a small planning module and limiting the prefix tokens to L=2. These choices appear to trade off performance for robustness, and the implications of increasing L remain unexplored.

During the discussion, the authors made significant efforts to address the reviewers' concerns and added new experiments that addressed some of the issues with the experiments (e.g., the addition of NLPO). However, the extensive discussions also revealed that the authors misunderstood key points about related work (L31-39, L128-131, and L47-49), and significant concerns about missing related work still remain. Due to the substantial amount of relevant approaches being overlooked, two reviewers and I believe the paper would require significant rewriting and reframing, which might also affect the paper's claims of novetly.

Another issue is that the authors neither updated the paper nor provided a unified response to all the reviewers explaining how they intended to address the major novelty concerns raised. This makes it difficult for me to recommend acceptance, as I would have preferred to read a unified explanation from the authors—ideally incorporated into updated introduction and related work sections—clarifying why the paper remains significant despite the related work overlooked in the original and only submission. Instead, I am left to piece together, from the various discussions, (1) the authors’ updated views on the paper's contribution and (2) their overall plan to address its shortcomings. This approach provides only a vague and incomplete picture of what we can reasonably expect in the camera-ready paper. Piecing this together is challenging because, e.g., the responses to novelty concerns were directed to two reviewers who had related yet distinct concerns.

Given all these uncertainties regarding the merits of the paper and how its shortcomings might be addressed, I recommend rejecting it.

**Additional Comments On Reviewer Discussion:**

The discussions revolved around the four weaknesses mentioned in related work. While some of them were addressed, Reviewers RkAj and diQR remain concerned about the novelty and the comparisons with the numerous previous works they listed in their reviews. In their view, the existence of related methods requires the paper to be significantly rewritten and reframed. Reviewer RkAj also expressed concerns about a technical limitation, specifically that only L=2 tokens are used to achieve 'planning.' These issues were highlighted again during the reviewer-AC discussion.

The concerns raised by Reviewers RkAj and diQR were brought multiple times to the attention of the other three reviewers, but the latter did not raise any objections to their arguments or concerns, which I believe strengthens the case for rejecting the paper.

---

### Decision · Program_Chairs · 2025-01-22

Reject